# *Staphylococcus* Infection: Relapsing Atopic Dermatitis and Microbial Restoration

**DOI:** 10.3390/antibiotics12020222

**Published:** 2023-01-20

**Authors:** John Hulme

**Affiliations:** Gachon Bio-Nano Institute, Gachon University, Seongnam-si 461-701, Republic of Korea; johnhulme21@gmail.com

**Keywords:** *Staphylococcus aureus*, infection, relapsing, atopic, dermatitis, restoration

## Abstract

Atopic Dermatitis (AD) skin is susceptible to *Staphylococcus aureus* (SA) infection, potentially exposing it to a plethora of toxins and virulent determinants, including Panton-Valentine leukocidin (PVL) (α-hemolysin (Hla) and phenol-soluble modulins (PSMs)), and superantigens. Depending on the degree of infection (superficial or invasive), clinical treatments may encompass permanganate (aq) and bleach solutions coupled with intravenous/oral antibiotics such as amoxicillin, vancomycin, doxycycline, clindamycin, daptomycin, telavancin, linezolid, or tigecycline. However, when the skin is significantly traumatized (sheathing of epidermal sections), an SA infection can rapidly ensue, impairing the immune system, and inducing local and systemic AD presentations in susceptible areas. Furthermore, when AD presents systemically, desensitization can be long (years) and intertwined with periods of relapse. In such circumstances, the identification of triggers (stress or infection) and severity of the flare need careful monitoring (preferably in real-time) so that tailored treatments targeting the underlying pathological mechanisms (SA toxins, elevated immunoglobulins, impaired healing) can be modified, permitting rapid resolution of symptoms.

## 1. Introduction

AD accounts for up 3–10% and 20% of skin diseases in adults and children. Although the link with processed foods remains unresolved [1], high-income countries are the most afflicted, with Swedish infants exhibiting a prevalence of up to 20% [2]. Depending on its severity and frequency, AD can dictate many aspects of the lives of sufferers and may be considered a disability akin to severe depression, cystic fibrosis, or renal disease. Exposure to a variety of everyday stressors (microbial infection (fungal and bacteria toxins), environmental pollutants (dioxins), heat, foods (acrylamides) [3,4], pressure, skin pH, and shock) can trigger AD [5]. Initial symptoms include pruritus and eczematous lesions in the knees, trunk, elbows, neck (flexural folds), and upper and lower extremities. AD can present as a chronic disease interspersed with lengthy periods of symptomatic resolution [6].

AD severity is evaluated using “The Scoring of Atopic Dermatitis (SCORAD)” and is based on the intensity of clinical signs and disease extent, with values > 50 considered moderate to severe. Comparable alternatives to SCORAD are the 5-point Investigator Global Assessment (IGA) scale and “The Eczema Area and Severity Index” (EASI), the former a modification of the Psoriasis and Severity Index (PASI) developed by Hanifin et al. [7]. All scoring systems are considered valid, with none of the methods demonstrating an advantage over the other [8]. Scoring and evaluation between patient and clinician can be done remotely [9] in real time [10] or using a variety of online or smartphone applications such as EZTrack for Atopic Dermatitis [11]. A recent validation study comparing the reliability of smartphone photographs taken and analyzed by 79 AD patients after software training (Imagine, LEO Innovation Lab, Copenhagen, Denmark) with five dermatologists showed high agreement and reliability in the assessment of mild to moderate AD [12]. Furthermore, automated severity scoring performed by deep neural networks exhibited > 90% accuracy providing further reassurance for dermatologists and patients alike [13].

The major medical co-morbidities associated with AD are microbial infections, specifically SA [14,15,16,17,18], affecting > 90% of patients. SA colonization is related to skin barrier disruption, a predominant Th2 immune response, elevated IgE levels specific to staphylococcal enterotoxin, and a reduction in skin microbial diversity. Other infections include viruses such as herpes simplex virus (HSV-1 or HSV-2) and coxsackie A6, which can result in eczema herpeticum (EH) and eczema coxsackium (EC). A summary of the major microbial pathogens associated with AD and their impact on the host immune response and skin microbiome is shown in Table 1. This review will briefly visit the various risk factors associated with SA infection, then, as a backdrop, utilize a patient case study to highlight the impact of associated toxins and their ability to impair healing and immune function, predisposing the subject to various triggers and AD flaring, further affirming previous correlations (infection severity) [6]. Finally, the review will close by looking at potential therapies that utilize the brain–skin–gut axis and their roles in restoring the skin microbiome and rebalancing the immune response regarding mild and severe AD.

## 2. Genetic Risk Factors and Epigenetic Regulators

A family history with inherited traits is the strongest risk factor for AD, with gene mapping studies accounting for 20% of the estimated heritability [19]. The term ‘heritability’ refers to the proportion of variation within a clinical feature that is attributable to genetic factors. Single null mutations in the gene encoding for the essential epidermal structural protein filaggrin (FLG) are currently the strongest genetic risk factor for AD, afflicting 10% of individuals of European ancestry, although 50% of those do not develop the disease [20]. In addition to the European mutations (R501X, 2282del4, R2447X, S3247X), copy number variations (intragenic repetitive FLG gene sequences) and the amount of filaggrin monomer expressed are additional risk factors for AD. Other candidate genes such as filaggrin-2 (FLG2) and those involved in the Th2 immune response (chromosomes 16p12.1–p11.2 (IL-4 receptor) and 5q31.1 (IL-13 receptor)) are thought to pose a significant risk as well [20]. FLG and FLG2, along with loricrin (LOR) and involucrin (IVL), are located on chromosome 1q21.3 [21,22,23,24]. Together with the S100A7 gene series, they constitute the bulk of the epidermal differentiation complex (EDC) [25]. Expression of the complex is internally modulated by cytokines (IL-4,13,17A,22) derived from T-helper cells (Th1,2,17,22). On the other hand, external modulation can occur via microorganisms (*Malassezia* spp. and *Staphylococcus epidermidis*) and numerous phytochemicals via the activation of the aryl hydrocarbon receptor (AHR). Prolonged AHR activation can result in accelerated keratinocytic differentiation, loss of sebocytes, and keratinous cyst formation [26]. Another modulator of keratinocyte differentiation is intracellular reactive oxygen species (ROS). ROS levels are kept in check by the oxidative stress-prone AHR and the antioxidative nuclear factor E2-related factor 2 (NRF2) systems. 

Keratinocytic differentiation is key to stratum corneum formation and epidermal [27,28] stratification. Their differentiation and subsequent development are regulated by the EDC genetic complex, signaling pathways, transcription factors, and epigenetic modifications. Epigenetic modifications can vary with exposure to acute and chronic environmental stressors leading to changes in genetic expression throughout all stages of keratinocytic development. These modifications include changes to the methylation status of DNA (methylation and hydroxymethylation), covalent additions or reductions to histone components (methyl-or acetyl groups), expression of noncoding RNAs (ncRNAs), nucleosome positioning, and ATP-dependent chromatin remodeling. Such changes are conducted via DNA methyltransferase (DNMT)1; histone deacetylases (HDAC)1/2; polycomb proteins (PcGs) Bmi1 and Ezh1/2; ATP remodelers Brg1 and Mi-2β of the chromodomain helicase DNA-binding (CHD) family; and histone demethylases (HDACs) Jmjd3 and Setd8, which together with several other transcription factors (in particular, p63) stimulate the proliferation and maturation of progenitor cells.

Of the many proteins, enzymes, and micro ribonucleic acids (miRNAs) involved in modulating genetic expression, the de/methylases and miRNA are the most prevalent. In a recently published [29] epigenome-wide association adult study, significant differences in methylation at 19 CpG sites and a partial correlation in altered gene transcript levels between the epidermis of healthy controls and epidermal lesions in AD patients were reported. Additional epigenetic studies have shown that high exposure to smoke can lead to hypomethylation of the thymic stromal lymphopoietin (TSLP) 5′CpG island in pregnant women and is positively correlated with AD [30]. Furthermore, tobacco smoke is thought to modify the Forkhead box-p3 (FOXP3) locus in umbilical cord blood and is associated with low Treg counts at birth. In addition to FOXP3, low Treg counts correlate with increased miR-223 expression. Other mRNAs that play a critical role in AD include (i) miR-124, which is downregulated in the lesional skin of patients with atopic eczema [31]; (ii) miR-143, a tumor suppressor that targets the IL-13Rα1 receptor reducing IL13 levels and inflammation in epidermal keratinocytes [32]; (iii) miR-146, reported to inhibit the expression of several proinflammatory factors, including IFN-γ and AD-associated genes CCL5, CCL8, and ubiquitin D (UBD) [33]; (iv) miR-155, which is positively correlated with AD severity and essential for the differentiation of Treg and T helper 7 (Th17) cells, and was recently shown to target protein kinase inhibitor α (PKIα) leading to its downregulation along with TSLP in a murine model [34]. TSLP promoter hypomethylation has been observed in keratinocytes isolated from AD patients resulting in the overexpression of TSLP in skin lesions [35]

The various epigenetic factors contributing to the polarized expression of EDC genes, interleukins, and cytokines can result in numerous functional deficiencies (decreased water retention, altered lipid formation, pH imbalance, ceramides, and sphingosine) involving proteins filaggrin, loricrin (LOR), involucrin (IVL), and desmosomes K1, K10, (envoplakin and periplakin) impairing corneocyte integrity [36,37]. Treatments have sought to address these deficiencies by either targeting inflammatory cytokines (monoclonal antibodies) or utilizing antioxidative AHR ligands and nuclear factor E2-related factor 2 (NRF2) activators (tapinarof, coal and soybean tar, and glyteer) to dampen reactive oxygen species (ROS) [38]. 

Finally, the epigenetic role of Staphylococcal enterotoxin B (SEB) is worth mentioning. In addition to the pore-forming toxins of *Helicobacter pylori, Listeria,* and *Salmonella* that repress a subset of immune genes (CXCL2, MKP2, and IFIT3) [39], SEB is adept at altering the methylation pattern of two gene regions (IKBKB and STAT-5B) both of which play a crucial role in T- cell maturation/activation [40].

### 2.1. Factors That Influence Staphylococcus Aureus Colonization of AD Skin

The reduced colonization resistance of AD skin, coupled with barrier deficiencies and a polarized Th2 immune response, renders the skin susceptible to SA biofilm formation and infiltration. The first stage in colonization is the covalent attachment of SA (sortase rich) to the skin, involving the recruitment of fibronectin, corneocytes, loricrin, and cytokeratin 10 [41]. Other factors that enhance colonization [42] and the potential for severe AD include IL-13 and IL-4, which down-regulate the expression of antimicrobial peptides (AMPs) such as human beta-defensin (HBD)-3 and cathelicidin (LL-37) [43,44]. Conversely, AMP expression is upregulated in the presence of coagulase-negative staphylococci (CoNS), *S. hominis* and *S. epidermidis*, which secrete phenol-soluble modulins (PSMs) γ and δ further amplifying AMP potency, thereby restricting in vitro growth of group A *Streptococcus* and *S. aureus* [45]. In addition to AMPs, other factors that aid colonization resistance include numerous acids (urocanic and pyrrolidone carboxylic) and lipids (including free fatty acid, ceramides, and sphingosine), filaggrin, and filaggrin degradation products (FDP) [46,47].

### 2.2. S. aureus Infection, Impairment of the Epidermal Barrier, and Potential Remodeling

When challenged, SA can secrete a battery of soluble receptor-mediated (α-toxin) and non-receptor-mediated (PSMs, δ-hemolysin) toxins, as well as enterotoxins A(SEA), SEB, and C(SEC), and a 22-kD superantigen, synonymous with toxic shock syndrome (TSS) [46,47,48,49]. In addition, the potency of said toxins is enhanced when delivered via extracellular vesicles (EV), which have been shown to play multiple roles in the development of AD and the activation of inflammatory macrophages. For example, mice subjected to repeated intranasal EV exposure induced a Th1 and Th17 cell response via Toll-like receptor 2 (TL2), resulting in neutrophilic pulmonary inflammation [50,51,52,53,54]. Many stressors (biochemical and physical) can induce SA and methicillin-resistant (MRSA) EV production, including iron depletion, sublethal concentrations of vancomycin or ampicillin, and temperature. The content of bacterial EVs has been shown to vary within a given temperature range (34–40 °C), with lower temperature SA EVs more cytotoxic to THP-1 cells (macrophages) than those secreted at 40 °C. Conversely, those secreted at 40 °C have been shown to contain virulence factors, primarily proteins and lipids, exhibiting amplified toxin-mediated erythrocyte lysis compared to their lower temperature counterparts [55]. 

Apart from toxins, other virulent factors include protein A coagulase, proteases, aureolysin, Staphylokinase, and collagen adhesins, which are collectively involved in repurposing the hosts’ proteins and the seeding of biofilms. In addition to SA, EVs are produced by other colonizing microbes (Malassezia species) and by the hosts’ repair mechanisms (keratinocytes), suggesting that EVs can modulate infection and AD inflammatory processes.

Impairment of the epidermal barrier occurs via various mechanisms involving numerous pore-forming toxins (Leukocidins ED, SF-PVL, AB, MF, HlgCB, HlgAB, and α-hemolysin). The most notable of these is α-hemolysin [56], which forms a heptameric β-barrel pore in the membranes of keratinocytes, initiating an ion imbalance. Cellular distribution is further enhanced with the binding of Protein A(PA) to tumor necrosis factor (TNF) receptors on the surface of keratinocytes triggering a sustained Th2 polarized inflammatory response via Langerhans cells. In the ensuing milieu, several other SA endogenous (kallikrein (KLK) 6, KLK13, and KLK14) proteases with a penchant for filaggrin and lipoproteins for the host’s fatty acids (linoleic acid (C18:2) [57] further amplify the Th2 inflammatory response by inducing thymic stromal lymphopoietin expression in a TLR2/TLR6-dependent manner in primary human keratinocytes [58]. Finally, SA becomes internalized by the keratinocytes inducing IL-1α production via Toll-like receptor 9 (TLR9). The various factors involved in SA adhesion and its subsequent internalization by keratinocytes are summarized in Figure 1 [59]. 

Other virulent factors include δ-Toxin and PSMα2 and PSMα3, the latter acting synergistically with IgE in the absence of bound antigen [60]. Last but not least is the role of superantigens (SAgs) and their impact on the corneocytes of the stratum corneum resulting in IgE antibody elevations following antigen presentation to Th2 cells [61]. SAgs can also trigger mast cell degranulation and induce Th2 cells to release IL-31. Children colonized with bacteria expressing SAgs have a higher disease severity than those with strains [62].

## 3. The Initial Impact of SA Toxins and AD Recurrence 

The impact of these toxins on a child’s skin is shown in Figure 2 [63]. At the time of record, the subject also exhibited (not shown) first systemic AD presentations in multiple cutaneous areas (neck, trunk, arms, and ankles). All AD presentations had dissipated within 12 h of administering antibiotics, and skin restoration was observed after 4 weeks. 

Without testing, it is debatable whether patient X had subclinical Th2 and Th22 cellular elevations in those areas’ prior infection. However, it seems likely that multiple pools of antigen-primed T and effector memory cells were generated during infection, which could be triggered later [62]. 

As discussed previously, only 5% with a genetic predisposition (European ancestry) for AD develop chronic disease. Unfortunately, patient X soon embarked on a long course of mild AD interspersed with periods of acute exacerbations (flares). The first of these resulted from a minor knee skin abrasion (no sign of infection), triggering a hypersensitive response in the folds of both arms coupled with intense itching < 12 h. Initial presentation appeared mixed (bacterial and fungi), with the administration of generic antibiotic creams (week 2) only serving to exacerbate. During the next two weeks, contact allergens were minimized by routinely (every morning) disinfecting bed linen (spraying) and bathing skin folds prior to sleep. The course of events is depicted in Figure 3, with asymptomatic resolution achieved in week 4. 

A similar case study involving a nine-year-old boy with chronic AD was reported by Kanchongkittiphon et al. [64], in which the total serum IgE level was markedly elevated, and the possibility of allergen (mites) immunotherapy was discussed. Regarding patient X, bleach baths were considered, but a targeted bathing approach concentrating on the folds, in combination with chloroxylenol spraying of bed linens (patient absent), was adopted, with time to resolution akin to generic treatments.

Further exacerbations were recorded at the end of 2017 (travel stress), followed by severe flaring after another fall in early 2018, suggesting post-traumatic stress involvement. The impact of psychological stress on AD severity, frequency of exacerbations, and reduced healing potential is well documented [65]. In such circumstances (Figure 3), the administration of selective serotonin reuptake inhibitors (SSRIs) sertraline and fluoxetine may have been considered, given the drug’s reported ability to reduce levels of anxiety, proinflammatory cytokines and IgE in AD patients [66,67,68,69,70]. In addition, SSRI paroxetine and fluvoxamine administered to AD patients were shown to reduce pruritus, suggesting central and peripheral nervous system involvement and the neurotransmitter 5-hydroxytryptamine (5-HT). However, anxiety is a complex disorder, and side effects can present if the underlying phenotype has not been identified [67,71]; consequently, recent investigations are currently restricted to topical SSRI usage, such as fluoxetine, which was shown to improve wound healing in mice by accelerating keratinocyte migration and rebalancing the local immune milieu [71]. The interplay between immune cells, neurotransmitters, and hormones and the effect on the levels of cortisol, adrenocorticotropic, and corticotropin-releasing hormones released by mature and immature keratinocytes and fibroblasts remains under investigation [72]. As a side note, for the first 24 months post-trauma (in the absence of an AD flare), the healing time for an abrasion remained significantly prolonged (10–14 days), suggesting IL-4 dysregulation. A model that probably best emulates that period might be the Interleukin-4 Transgenic Mouse developed by Zhao et al. [73] regarding aberrant wound healing. 

Lengthy and frequent AD recurrences can lead to observable thickening in the flexural folds with papule-nodule transitions, plaques with excoriation, lichenification, and significant remodeling (Figure 4A–C). Changes in epidermal thickness resulting from chronic AD are usually accompanied by a higher density of nerve fibers [74], Th2, and epidermal dendritic cells, increasing regional susceptibility to additional environmental factors such as heat, pressure, and sweat, the latter often associated with soft wound infections. Soft wound treatments range from cold atmospheric pressure plasma [75], silver and zinc nanoparticles, platelet therapies, chitin bandages, and tens of others. The scarring and nodulation resulting from AD recurrence are shown in Figure 4C.

During 2018–2021, patient X relapsed multiple times, putting them at significant risk from the so-called atopic march that can lead to asthma and allergic rhinitis in later life. However, nodulations rooted in the dermis and epidermis did show significant shrinkage following topical povidone administration in early 2022. Moreover, additional wounds or abrasions rapidly treated with povidone (PV) later in the same year did not evoke flaring and healed quicker, which may have been due to limited IgE production leading to reductions in reagin antibody titers or how the treatment (psychological) was administered [76]. Interestingly, a recent pilot phase II study investigating the combinatorial antiseptic and anti-inflammatory action of liposomal polyvinyl-pyrrolidone (PVP)-iodine gel reduced the global clinical severity score (GCSS) for all dermatoses [77,78]. Moreover, PV preventions have been shown to inhibit *C. albicans,* and SA biofilm formations often encountered with chronic AD [79]. Side effects included itching and burning (9 and 14%). Povidone has been shown to counter various SA and MRSA armaments (α-hemolysin, phospholipase C, lipase, elastase, and β-glucuronidase) [80] and remains the gold standard for biocides in hospital settings [81,82]. A timeline documenting the progression of AD and associated triggers regarding patient X can be found in Table 2.

However, the misuse of biocides can lead to a surge in MRSA cases (cross-resistance), as reported during the COVID-19 pandemic [83]; thus, alternatives treatments need to be developed to limit the emergence of multi-drug resistant microbes (fungi, bacteria, and viruses) [84]. A recent review by Bieber et al. documents a number of approved inhibitor therapies (IL-4Rα “dupilumab”, anti-IL-13 “tralokinumab”, and JAK1/2) and those currently in clinical trials [85], which might have been a better option (dupilumab) for patient X. 

## 4. Restoration of Microbial Homeostasis

The skin microbiome consists of regionally specific microbial communities hosting over 1000 species from 19 phyla. These communities’ microbial composition depends on on-site skin physiology, and the microenvironments (dry, moist, foot, sebaceous) encountered there (Figure 5). For example, where water loss is relatively high (dry) and follicle density (inner forearm) low, fungal species (*Malassezia*) predominate; in moist regions (folds), bacteria species (*Staphylococcus* and *Corynebacterium*) hold sway and where follicle density and skin surface area are highest (Sebaceous) (head, neck, and chest) lipophilic Propionibacterium species dominate. Finally, various fungal (*Aspergillus, Cryptococcus*, *Rhodotorula*, *Malassezia,* and *Epicoccum*) species can be found in the feet [86].

Collectively, these microbially diverse communities constitute the bulk of the skin microbiome. The density and diversity of these communities vary with age, sex, and race of the host. Interactions (communications) with the host occur via epithelial and immune cells and are mutually beneficial with the community’s providing protection from pathogenic or translocated species and the host offering “food and board.”

However, these interactions can be perturbed by environmental conditions (skin infection or trauma), providing opportunities for other species, such as SA, that are adept at prioritizing overgrowth and biofilm formations [86,87], leading to prolonged barrier and immunological dysfunction. Of the many SA strains residing on the skin, the clonal complex 1 (CC1) takes precedence during severe AD, whereas CC30 is prominent on the skin of asymptomatic AD individuals. 

Numerous therapeutic approaches, including endolysins, bacteriocins (Nisin Z, Lacticin 3147), small molecule inhibitors (2-aminoimidazole), bacteriophages, and nanoparticles or combinations thereof, can be used to prevent SA biofilm formation [80,88]. However, in the case of patient X, concerns extended beyond the skin, as the longevity and frequency of the disease suggested the initial involvement of the gut (microbiome) and, later on, the brain (stress) via the gut–brain–skin axis [89]. Animal model (murine) research regarding factors that affect the axis has revealed the immunoregulatory importance of dietary AHR ligands and hydrophilic bile acids. For example, mice fed an indole-3-Carbinol(I3C)-supplemented diet showed better monocyte-to-dendritic skin differentiation than controls [90], which is consistent with previous reports highlighting the shift from macrophages via the modulation of transcription factors Irf4 and Blimp-1. Moreover, I3C diet supplementation reversed antibiotic-treated mice with impaired differentiation, suggesting a role for microbial-derived AHR ligands [90]. The effects of dietary tryptophan-free deficiencies on an experimental autoimmune encephalomyelitis (EAE) model resulted in higher disease scores, increased expression of pro-neuroinflammatory molecules (Ccl2, Nos2, and Tnfa), and prolonged [91] recovery in mice. 

The catabolism of dietary tryptophan represents a major source of indoles for humans. A recent clinical study comparing elevated fasting serum and normal IgE levels in infant AD cohorts showed the former had increased serum indole-3-acetic acid (tryptophan metabolite) and tryptophan levels [92], suggesting impaired tryptophan metabolism. Moreover, the cohort with normal IgE levels also showed a significant decrease in primary conjugated bile acids (taurocholic, taurochenodeoxycholic acid, and glycochenodeoxycholic). In contrast, the high IgE AD cohort showed cholic acid and chenodeoxycholic acid elevations, suggesting reduced conjugation with a potential risk of *E. coli* intestinal overgrowth [93].

Of the many indole derivatives currently under investigation [94,95,96,97], the work by Yu et al. demonstrated the potential of indole-3-aldehyde treatments for lesional and non-lesional AD patients by attenuating skin inflammation and TSLP keratinocyte expression in an AD-like dermatitis murine model (MC903) [98]. More recently, Fang et al. further emphasized the importance of microbially (*Limosilactobacillus reuteri*) produced indole derivatives (indole lactic acid and indole propionic acid) in attenuating AD by demonstrating its ability to suppress IgE, TSLP, IL-4 and 5 levels in mice [99]. Regarding the anti-inflammatory potential of hydrophilic bile acids (tauroursodeoxycholic acid (TUDCA)), reports have shown that they reduce the unfolded protein response (UPR) seen in keratinocytes [59,100], hypersensitivity in asthmatic patients, as well as reverse the amyloid beta-oxidation of mitochondrial membranes in Alzheimer’s murine models [101,102]. Whether TUDCA alone or blended bile acids treatments can re-establish skin microbial homeostasis directly or indirectly via changes in the gut microbiota (probiotic interventions or fecal matter transfer) requires further investigation. In addition to indole derivatives, a study by Hwang et al. investigated whether n-3 polyunsaturated fatty acids (PUFA) in red blood cells (RBC) had a protective effect on the atopy of 380 preschoolers. Preschoolers with atopy had lower total RBC n-3 PUFA levels and a greater n-6/n-3 PUFA ratio than the controls, suggesting lower n-3 PUFA may play a role in child atropy [103].

Microbial analysis of AD fecal matter by Reddel [104] and co-workers demonstrated that microbiomes of 19 AD children were characterized by an increase of *Faecalibacterium*, *Sutterella*, *Oscillospira*, Bacteroides, and Parabacteroides and a significant reduction in short-chain fatty acid (SCFA)-producing bacteria (*Bifidobacterium*, *Blautia*, *Coprococcus*, *Eubacterium*, and *Propionibacterium*) compared with 18 healthy individuals. More recently [105], Kim et al. showed that fecal matter transfer (FMT) from a healthy murine donor ameliorated AD in mice; resulted in reduced IgE levels and the numbers of mast cells, eosinophils, and basophils; and the restoration of the Th1/Th2 balance. An initial clinical study (*n* = 9) [106] assessing the clinical efficacy of capsule FMT as a viable treatment in adults with moderate-to-severe AD over eight months demonstrated a significant reduction in the average SCORAD score from baseline pre-FMT as well as a decrease in corticosteroid usage, disease severity, and partial assimilation of the donor’s microbiota.

## 5. Conclusions

AD is a complex, multifaceted disease with varying degrees of severity (mild, moderate, and severe) and recurrence. Some reports have speculated that a comprehensive understanding of the skin’s ability via microbial modulation to resist SA colonization would reduce AD severity and recurrence, leading to better therapeutics. However, personalized modulation is still in its infancy as the skin virome is yet to be typed, and the ability of species to limit or enhance SA virulence is yet to be elucidated [107,108]. Regarding patient X, a unique set of conditions was presented that allowed for AD monitoring and the changing factors (trauma-stress-temperature and pressure) that initiated flaring over the course of the disease. After numerous therapeutic failures and temporary successes spanning five years, the link between initiation and flaring was finally broken with the topical administration of the biocide povidone. However, the use of biocides has a detrimental effect on commensal microbiota. Thus, alternative treatments that rebalance or partially restore the skin microbiome via bacterial species, such as *Bifidobacterium longum* or bile acids (TUDCA), that better mediate tryptophan metabolism via the gut–brain axis can be employed regarding mild or non-lesional AD. Concerning chronic or severe AD, in which the dysregulation of the gut immune system is suspected, FMT or partial FMT may be adopted to restore the Th1/Th2 balance and reduce serum IgE levels preventing atopic march [105].

## Figures and Tables

**Figure 1 antibiotics-12-00222-f001:**
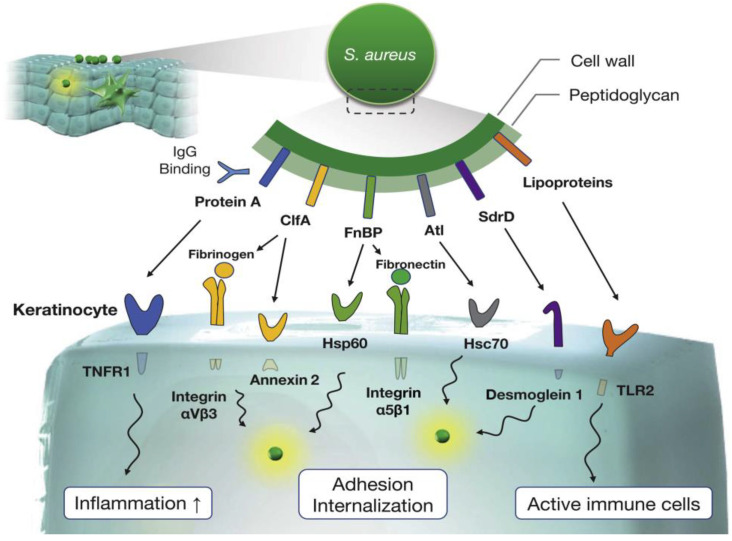
Interactions between SA cell wall proteins and keratinocytes. Cell wall proteins, clumping factor A (ClfA), fibronectin-binding proteins (FnBP), autolysin (Atl), and serine aspartate repeat-containing protein D(SdrD) involved in adhesion and keratinocyte internalization via respective receptors (heat shock protein 60 (Hsp69), Integrinα5β1, Annexin2, with fibronectin bridging, Desmoglein1, and Heat shock cognate 71 kDa protein (Hsc70)). In addition, PA induces inflammation by binding to tumor necrosis factor receptor-1 (TNFR1), and Toll-like receptor 2 (TLR2) senses lipoprotein following the activation of immune cells. Reproduced with permission from Elsevier [59].

**Figure 2 antibiotics-12-00222-f002:**
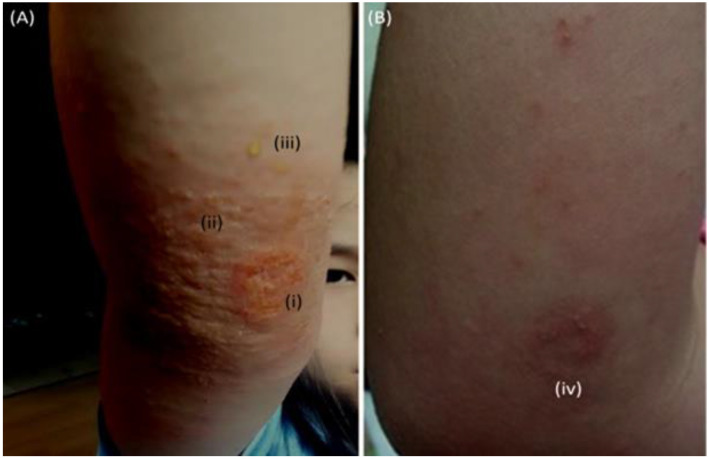
Photograph of SA-infected abrasion (patient X (Age 9-female); 4 October 2016) following administration (1 October 2016) of low-dose erythromycin and corticosteroids (**A**). Permanganate (aq) cleansing, followed by administration of high-strength oral antibiotics (amoxicillin) (5 October 2016); overnight resolution of non-localized symptoms (neck, head, and trunk) highest recorded patient temperature 38.5 °C. Day 30 barrier restored (**B**). (i) Infected area, (ii) immunological reaction to SA toxins, (iii) honey-colored fluid, (iv) skin repaired, sporadic folliculitis. Modified with permission from Springer [63].

**Figure 3 antibiotics-12-00222-f003:**
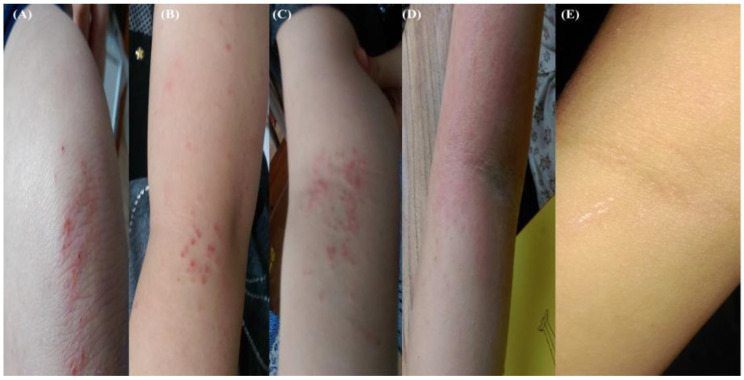
Suspected stress-induced flare without infection. Photograph (10 October 2017) of grazed knee, healing time 10 days (**A**). Photograph (11 October 2017): flaring, outer right (**B**) and inner left forearms (**C**). Photograph (26 October 2017): severe inflammation, Fucidin and mupirocin treatments discontinued, bacterial and fungal (*Candida* and *Malassezia*) involvement suspected (**D**). Resolution of AD symptoms following daily disinfection (1% Chloroxylenol spraying) of bedding covers (**E**).

**Figure 4 antibiotics-12-00222-f004:**
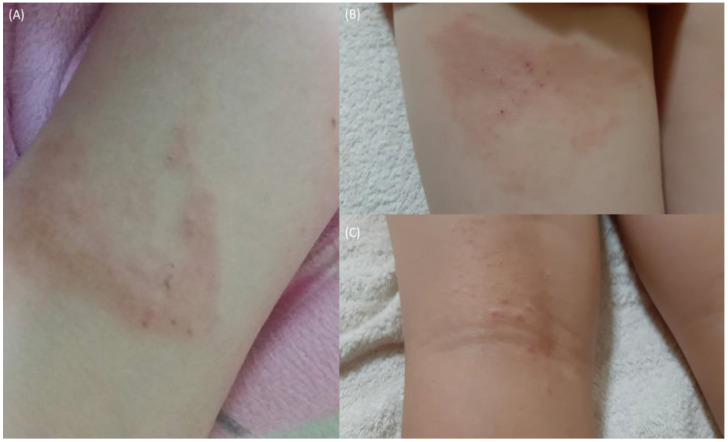
Recurrent AD flaring. (**A**) (16 March 2019) heat and pressure-triggered flare (posterior left knee). (**B**) (16 June 2021) heat and pressure-triggered flare (posterior left thigh). (**C**) (16 June 2021) scarring and nodulation (posterior left knee) following repeated flaring.

**Figure 5 antibiotics-12-00222-f005:**
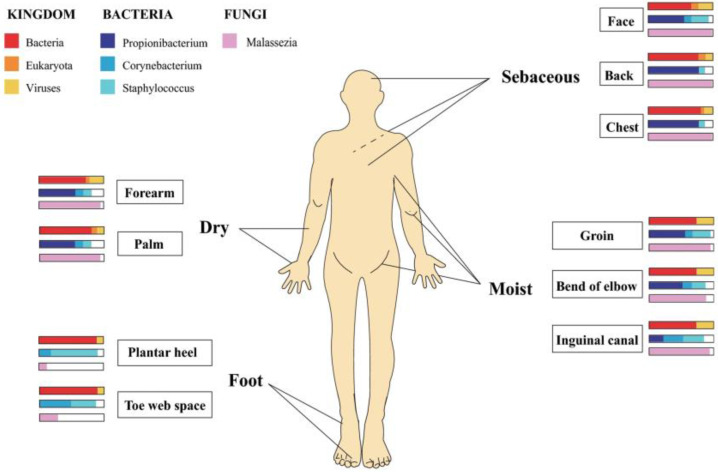
Regional distribution of microorganisms in human skin. Dry (forearm and palm), sebaceous (head, back, and chest), moist (groin, bend of elbow, and inguinal canal), and foot (plantar heel and toe web space). Wide and narrow bar charts represent the anatomical location and relative abundance of microorganisms distributed in those areas. Reproduced with permission from Springer [87].

**Table 1 antibiotics-12-00222-t001:** Microbial skin infections and resultant changes in immune function and microbial composition associated with AD.

Infectious Species	Clinical Features	Immune Dysfunction	Microbiome	Reference
*S. aureus* and Methicillin-resistant *Staphylococcus aureus* (MRSA)	Weeping, honey-colored crusts and pustules, both interfollicular and follicular-based (folliculitis) Abscesses, fever, and lymphadenopathy	↓Antimicrobial peptides ↑IL-13, IL-4 B-cell Ig class switching to IgE ↑type 2–related chemokines (CCL13, CCL17, CCL18, and CCL22) ↑Degradation of immunoglobulin G (IgG)	↓ coagulase-negative *Staphylococci* (CoNS) (*S. epidermidis*, *S. hominis,* and *S. lugdunensis*) ↑ *S. aureus*	[14,15]
Beta-hemolytic streptococcal	Bright red erythema, thick-walled pustules, and heavy crusting	↑Degradation of IgA, IgM, IgD, and IgE	↑ *S. aureus*	[16]
Herpes simplex virus (HSV) molluscum contagiosum (MC), eczema vaccinia (EV), and eczema coxsackium (EC)	Superficial clusters of dome-shaped vesicles and/or small, round, punched-out erosions	↑IL-13 and IL-4 ↓ IFN-γ and TNF-α	↑ *S. aureus*	[17]
*Malassezia globosa* and *Malassezia restricta* nanovesicles	Pruritic monomorphous papules and/or pustules. Hypo- or hyper-pigmented non-inflammatory lesions	↑ IgE↑ auto-reactive T cells induces autoreactivity to human proteins	*↓S. aureus*	[18]

**Table 2 antibiotics-12-00222-t002:** Changes in the SCORAD value of patient X in response to various triggers.

Date & Location	Trigger	Morphological Description	SCORAD	Treatment	Time to Resolution	Reference
4 October 2016 A*	Fall and SA-infected abrasion	Marked erythema (deep or bright red), papulation; disease is widespread in extent	56.87	Permanganate (aq) cleansing high-strength oral antibiotics (amoxicillin)	30 days	Figure 2A,B [63]
26 October 2017 B*	Fall and abrasion	Perceptible erythema clearly perceptible induration/papulation	43.7	1% Chloroxylenol bathing prior sleep	16 days	Figure 3A–E
27 December 2017 B*	Stress	Slight but definite erythema (pink), slight but definite induration	19.45	1% Chloroxylenol bathing prior sleep	12 days	N/A
27 January 2018 A*	Fall and abrasion	Perceptible erythema induration/papulation	26.9	1% Chloroxylenol bathing prior sleep	16 days	N/A
16 March 2019 A*	Heat and pressure	Perceptible scarring and Skin thickening (lichenification), itching	27.8	UV-B and exercise	14 days	Figure 4A
16 June 2021 C*	Heat and pressure	Perceptible scarring, persistent nodulation, itching Skin thickening (lichenification)	21.8	UV-B and exercise	10 days	Figure 4B,C
10 March 2022 C*	Foot wound	No inflammatory signs of local or systemic atopic dermatitis; nodulation and occasional itching	7.4	Topical application of Povidone	5 days	N/A

SCORAD values were assessed using photographs (Figure 2, Figure 3 and Figure 4) and at A* (Eulji University Hospital Daejeon South Korea), B* (Royal Manchester Children’s Hospital UK), C*, and at home by guardians utilizing online software (https://scorad.corti.li/ accessed on 16 March 2019). Permission for other photographs for 27 December 2017, 27 January 2018, and 10 March 2022 was not granted (N/A).

## Data Availability

Not applicable.

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
