# Peer review of "Staphylococcus* Infection: Relapsing Atopic Dermatitis and Microbial Restoration"

_antibiotics, 2023, doi:10.3390/antibiotics12020222_

Round 1

Reviewer 1 Report

I consider the topic original and relevant in this field, all the references are appropriate , but maybe some few more could be added to the list. Conclusions are consistent and concise redacted and the arguments well presented.

Author Response

Thank you for your consideration

English has been checked 

Reviewer 2 Report

Dear Editor,

I have reviewed the review article entitled "Staphylococcus Infection: Relapsing Atopic Dermatitis and Microbial Restoration" by Hulme JP. The review is well designed and well written.

The whole manuscript should be revised for typing mistakes

Author Response

The manuscript has been checked for typos

Reviewer 3 Report

General comments

The Review entitled “Staphylococcus Infection: Relapsing Atopic Dermatitis and Microbial Restoration” provides a wealth of information related to a complex subject. However, there are issues concerning the integrated presentation of the information. More significantly the review centers around the singe case of “patient X” as an example of restoration but without statistical references to other cases or clear references where the data for patient X are taken from. Also, the results from patient X are not presented in a concise table to give an overview of the case and a greater general message.

A table (or tables) presenting different types/severities of atopic dermatitis related to microbial infections, immune response and treatments would be highly desirable. Such tables are currently missing while the continuous discussion in the text, does not help in presenting the greater picture which is desirable for a review article. The reader is entangled in a maze of information without a distinct punge/overview.

Other

Many abbreviations are included prior to their presentation as a whole word (e.g., line 58 “SA”) or abbreviations without explanation (e.g., line 285 “FMT”) or abbreviations used for a few times (e.g., line 41: filaggrin (FLG)”).

Line 127, paragraph 3: “3. Infection and AD recurrences (patient X)”. Please refer systematically where you got the presented data and photos. Also in figures 2, 3, 4 and supplement. Is reference [39] the source of all information concerning patient X? Please state clearly what are newly presented data and what are data from bibliography.

Lines 49-52: “On the other hand, said genes are downregulated by T-helper cells (TH1,2,17,22)-derived inflammatory cytokines (IL-4, 13,17A,22), causing functional deficiencies such as decreased water retention, altered lipid formation and impaired corneocyte integrity [13,14]”. Please improve sentence.

Line 63: “cytokeratin10”, line 216 “there(figure5)”. Please inset gap.

Lines 123-124: Th2 or Th-2 cells?

Line 329 “Escherichia coli” and “E. coli”. Please write in italics.

Lines 222-223 “In homeostasis, these communities interact with the skin in a reciprocal manner by sponsoring host immune tolerance and enhanced barrier function in exchange for residence and useful metabolic products.”. Please clarify sentence.

Lines 264-7 “Reports regarding improved gut barrier function due to regulated AHR activation, CD4+ T-cell differentiation, enhanced IL-22 and IL-10 expression, and resultant elevations in antimicrobial peptides following exposure to microbial-produced indole derivatives are well known [68-71]”. Please improve sentence.

Line 285, define FMT.

The photo in supplement could be included in the text.ld be included in the text.

Author Response

Hello

Please refer to the attached

file 

Thank you

Round 2

Reviewer 3 Report

General

This is an improved version of the previous manuscript. An impression of a lack of cohesiveness remains.

Specific

Your Abstract needs a better ending. You must summarize better what you wrote in text (e.g., FMT / TUDCA) and give an outlook for future treatments (e.g., fecal transfer).

Still some mistakes remain concerning the names of strains. Write the genus and species of every strain in italics (e.g., line 235). Use small letter after the genus name (e.g., line 137, 444, 447 etc.).

Please insert gaps between words and parentheses (e.g., lines, 7, 40, 76, 244, 306 to mark a few…), numbers and units (e.g., lines 225, etc.), words (e.g., line 80 “Th1,2,17,22”).

Line 22 “3-10 % and 20%”: decide if you place a gap between % and numbers. We suggest that you place a gap between all numbers and %.

Line 205 “The impact of these toxins on the skin of a child is shown in figure 2”.” Insert appropriate refence.

Insert references for figures 3 and 4.

Author Response

Gel or flow

Addressed at the end of the introduction

This review will briefly visit the various risk factors associated with SA infection, then, as a backdrop, utilize a patient case study to highlight the impact of associated toxins and their ability to impair healing and immune function, predisposing the subject to various triggers and AD flaring, further affirming previous correlations (infection severity) [6]. Finally, the review will close by looking at potential therapies that utilize the brain-skin-gut axis and their roles in restoring the skin microbiome and rebalancing the immune response regarding mild and severe AD.

Specific

Your Abstract needs a better ending.

Furthermore, when AD presents systemically, desensitization can be long (years) and intertwined with periods of relapse. In such circumstances, the identification of triggers (stress or infection) and severity of the flare need careful monitoring (preferably in real-time) so that tailored treatments targeting the underlying pathological mechanisms (SA toxins, elevated immunoglobulins, impaired healing)  can be modified, permitting rapid resolution of symptoms

You must summarize better what you wrote in text (e.g., FMT / TUDCA) and give an outlook for future treatments (e.g., fecal transfer).

Conclusion

However, the use of biocides has a detrimental effect on commensal microbiota. Thus, alternative treatments that rebalance or partially restore the skin microbiome via bacterial species, such as Bifidobacterium longum or bile acids (TUDCA), that better mediate tryptophan metabolism via the gut-brain axis can be employed regarding mild or non-lesional AD. Concerning chronic or severe AD, in which the dysregulation of the gut immune system is suspected, FMT or partial FMT may be adopted to restore the Th1/Th2 balance and reduce serum IgE levels preventing atopic march [105].

Still some mistakes remain concerning the names of strains. Write the genus and species of every strain in italics (e.g., line 235). Use small letter after the genus name (e.g., line 137, 444, 447 etc.).

Please insert gaps between words and parentheses (e.g., lines, 7, 40, 76, 244, 306 to mark a few…), numbers and units (e.g., lines 225, etc.), words (e.g., line 80 “Th1,2,17,22”).

Line 22 “3-10 % and 20%”: decide if you place a gap between % and numbers. We suggest that you place a gap between all numbers and %.

Line 205 “The impact of these toxins on the skin of a child is shown in figure 2”.” Insert appropriate reference. Done with additional corrections

Insert references for figures 3 and 4. This is new data